# Global Transcriptomic Analysis of Zebrafish Glucagon Receptor Mutant Reveals Its Regulated Metabolic Network

**DOI:** 10.3390/ijms21030724

**Published:** 2020-01-22

**Authors:** Qi Kang, Mengyi Hu, Jianxin Jia, Xuanxuan Bai, Chengdong Liu, Zhiqiang Wu, Wenbiao Chen, Mingyu Li

**Affiliations:** 1School of Pharmaceutical Sciences, Fujian Provincial Key Laboratory of Innovative Drug Target Research, Xiamen University, Xiamen 361102, China; kangqix@126.com (Q.K.); humyi018@163.com (M.H.); jia_jianxin@126.com (J.J.); bxxgxfcsssy@126.com (X.B.); 2State Key Laboratory of Cellular Stress Biology, School of Life Sciences, Xiamen University, Xiamen 361102, China; 3The Key Laboratory of Mariculture, Education Ministry of China and College of Fisheries, Ocean University of China, Qingdao 266003, China; liuchengdong@ouc.edu.cn; 4School of Marine Life Sciences, Ocean University of China, Qingdao 266003, China; wuzhiqiang@ouc.edu.cn; 5Department of Molecular Physiology and Biophysics, Vanderbilt University School of Medicine, Nashville, TN 37232, USA; wenbiao.chen@vanderbilt.edu

**Keywords:** glucagon receptor, zebrafish, RNA sequencing, metabolic network, diabetes

## Abstract

The glucagon receptor (GCGR) is a G-protein-coupled receptor (GPCR) that mediates the activity of glucagon. Disruption of GCGR results in many metabolic alterations, including increased glucose tolerance, decreased adiposity, hypoglycemia, and pancreatic α-cell hyperplasia. To better understand the global transcriptomic changes resulting from GCGR deficiency, we performed whole-organism RNA sequencing analysis in wild type and *gcgr*-deficient zebrafish. We found that the expression of 1645 genes changes more than two-fold among mutants. Most of these genes are related to metabolism of carbohydrates, lipids, and amino acids. Genes related to fatty acid β-oxidation, amino acid catabolism, and ureagenesis are often downregulated. Among *gcrgr*-deficient zebrafish, we experimentally confirmed increases in lipid accumulation in the liver and whole-body glucose uptake, as well as a modest decrease in total amino acid content. These results provide new information about the global metabolic network that GCGR signaling regulates in addition to a better understanding of the receptor’s physiological functions.

## 1. Introduction

Glucagon, derived from pancreatic α-cells, activates glucagon receptor (GCGR) signaling to increase hepatic glucose production and maintain normal blood glucose levels during fasting [1]. GCGR is a seven-transmembrane class B G-protein-coupled receptor [2,3]. Stimulation of GCGR activates adenylyl cyclase and increases cAMP levels [4]. Elevated cAMP stimulates several pathways, resulting in an increase of gluconeogenesis, glycogenolysis, and fatty acid oxidation [5]. Inhibition or inactivation of GCGR is shown to lower blood glucose in preclinical models of type 1 and type 2 diabetes [6,7,8]. Consequently, GCGR antagonists and neutralizing antibodies have emerged as a class of novel therapeutics for treating diabetes [9,10,11,12]. Several antagonists and antibodies are undergoing different stages of clinical trials, but none have been approved for clinical use [13].

Beyond suppressing hyperglycemia, disruption of GCGR also leads to increased insulin sensitivity, hypoglycemia, hyperglucagonemia, hyperaminoacidemia, increased plasma LDL, increased GLP-1 and FGF21 levels, decreased adiposity, and hyperplasia of pancreatic α-cells in mouse models [6,14,15,16]. The disruption of GCGR by antagonism or gene knockout in animal models also causes dysregulation of other metabolic processes, including cholesterol absorption, fatty acid utilization, white adipose tissue browning, and bile acid metabolism [17,18,19,20]. In humans, patients with GCGR mutations develop a syndrome known as Mahvash disease, characterized by hypercalcemia, hyperglucagonemia, and α-cell hyperplasia [21,22,23,24]. These studies from animal models and human patients suggest that GCGR regulates a network of signaling pathways in organismal physiology.

We previously generated GCGR knockout zebrafish that display hypoglycemia, hyperglucagonemia, and compensatory α-cell hyperplasia, demonstrating that the function of glucagon signaling is at least partially conserved between humans and zebrafish [25]. The amenability of zebrafish to live imaging, chemical screening, and genetic manipulations should facilitate deeper mechanistic understanding of GCGR functions and α-cell biology using this mutant line. Moreover, since it is hard to systemically dissect the genes network of GCGR in rodent models, we took advantage of zebrafish model, and performed RNA sequencing (RNA-seq) analysis of whole fish to provide a comprehensive view of its global transcriptomic changes in the absence of GCGR signaling. 

## 2. Results

### 2.1. The Transcriptome of Wildtype and Gcgr^−/−^ Larvae

To survey the global transcriptomic changes of *gcgr^−/−^* mutants, we extracted total RNA from three biological replicates of both wildtype and *gcgr^−/−^* samples and performed high-throughput RNA-seq at seven days post fertilization (dpf). Together, these samples generated 21.94 million (M) pairs of raw reads, with a clean read ratio greater than 99.7%. After the quality filtering, clean reads were mapped to the zebrafish genome (GRCz11) using HISAT (v2.0.4) [26]. Among these reads, 89.55–90.55% clean reads were successfully mapped to the zebrafish genome, and the number of uniquely mapped reads was 63.98–65.53%. Then, clean reads were mapped to a reference transcriptome of zebrafish using Bowtie2 [27], and total 25,955 genes were detected. The average genes mapping ratio was 79.68% (79.28–80.03%), with genes uniquely mapped between 69.0%–69.21% (Table 1). Principal component analysis (PCA) demonstrated that the *gcgr*-mutant datasets clustered distinctly from wildtype control (Figure 1A).

### 2.2. Gene Ontology (GO) Enrichment Analysis of Differentially Expressed Genes

A total of 24,831, 24,832, and 24,889 genes were detected with more than one read in WT1, WT2, and WT3, respectively, while 24,666, 24,629, 24,655 genes were detected in *gcgr^−/−^*1, *gcgr^−/−^*2, *gcgr^−/−^*3, respectively. Comparison of RNA-seq data between the two genotypes identified by 1645 DEGs, with 437 upregulated and 1208 downregulated in *gcgr^−/−^* mutants (Figure 1B and Appendix A). All the DEGs were annotated with gene ontology (GO) terms, and sorted into three major functional categories: “biological process”, “cellular component,” and “molecular function” (Figure 1C–E). Cellular process (22%), metabolic process (13%), and biological regulation (12%) were the main subcategories of the biological process group. Under the category of “cellular component,” “cell” (25%) was the largest class, followed by “membrane” (16%), and “organelle” (16%). As for “molecular function”, “binding” (47%) and “catalytic activity” (30%) were the major classes. These results of GO enrichment indicate that multiple biological processes changed in the *gcgr*-deficient fish. Since many functional categories were affected, we seek to determine whether there was any behavior or development difference between wildtype and *gcgr^−/−^* mutant. However, no significant difference in swimming speed and distance between mutant fish and wildtype fish were observed (Figure 2). Moreover, there were no obvious defects in the development of *gcgr^−/−^* mutants compared to wildtype specimens (Appendix A)

### 2.3. Kyoto Encyclopedia of Genes and Genomes (KEGG) Analysis Reveals that GCGR Deficiency Alters Multiple Metabolic Pathways

To better understand the GO-annotated DEGs in the zebrafish *gcgr^−/−^* mutant larvae, DEGs were subjected to the KEGG database for canonical signaling pathway analysis. The 1645 DEGs were significantly enriched in 44 different signaling pathways, and these pathways were most related to metabolism (12), human disease (11), organismal systems (10), cellular processes (4), genetic information processing (4), and environmental information processing (3).

Strikingly, we found that many genes involved in several metabolic processes were notably affected. Since GCGR is important for the regulation of metabolism, we then further analyzed these pathways. Interestingly, in metabolism pathways, the downregulated genes are far more than the upregulated genes. These data suggest that the GCGR knockout disrupts multiple metabolic pathways in zebrafish (Figure 3).

### 2.4. GCGR Regulates Lipid Metabolism

For lipid metabolism, 52 genes were altered with 41 downregulated and 11 upregulated (Figure 4A and Appendix A). Among these genes, there were 12 genes in fatty acid related metabolism pathways, including *fads2* (log2 values, −1.78), *lrfn4b* (−1.65), *fasn-like* (−1.49), *ppt2-like* (−1.45), *acoxl* (−1.40), *acaa2-like* (−1.27), *hadhaa* (−1.27), *eci1* (−1.16), *aldh2.2* (−1.11), *elovl4b* (−1.02), *tecrl2b* (−1.09), *elovl7b* (1.95). These genes are involved in the pathways of fatty acid metabolism (ko01212), fatty acid biosynthesis (ko00061), fatty acid elongation (ko00062) and fatty acid degradation (ko00071), and biosynthesis of unsaturated fatty acid (ko01040). These pathways were all affected, suggesting that global GCGR deficiency resulted in impaired fatty acid metabolism (Figure 4 and Appendix A). In the cholesterol metabolism (ko04979) pathway, *apobb.2* (−1.63), *ptchd3* (−1.61), *cyp27a1.4* (−1.34), *cyp27a7* (−1.05), and *pcsk9* (−1.11) were decreased, while *abca1-like* (4.59), *ldlr* (4.53), *lipca* (2.53), *poa4-like* (2.32), *pltp* (1.22), and *soat2* (1.08) were increased. These data suggest that knockout of zebrafish GCGR disrupted cholesterol metabolism. In the sphingolipid metabolism pathway (ko00600), *arsa-like* (−4.92), *prr13* (−1.37), and *neu3.4* (−1.29) were all decreased. In the glycerophospholipid metabolism pathway (ko00564), *phospho1*(−1.03), *pla2g4c* (−3.09), *pcyt1-like* (−1.15), and *mfsd10* (−1.15) were decreased, while *pla2g4f-like* (2.15) and *plpp4* (1.42) were increased.

Glucagon regulates lipid metabolism through p38 MAPK-, PPARa-, and FGF21-dependent mechanisms [20,28]. Many genes in MAPK and PPAR pathway were indeed enriched in our study. Fourteen DEGs were enriched in PPAR signaling pathway, which included *isg15* (−4.01), *trim39-like* (−2.33), *fabp11b* (−2.19), *ubc-like* (−2.12), *cyp8b3* (−1.83), *fads2* (−1.78), *samd3-like* (−1.50), *acoxl* (−1.40), *cyp27a1.4* (−1.34), *fabp1b.1* (−1.12), *fabp7a* (−1.11), *cyp27a7* (−1.05), *plin1-like* (−1.02), and *pltp* (1.22). Most of them are the target of the PPARα, PPARβ, or PPARγ in peripheral tissues (Figure 4B). Except for *pltp*, these genes were all downregulated, suggesting a downregulation of PPAR activity. For MAPK signaling pathway, *pla2g4c* (−3.09), *casq1a* (−1.49), *flt3* (−1.48), *angpt2-like* (−1.37), *stmn1a* (−1.33), *xaf1* (−1.24), *fgf23* (−1.22), *gna12* (−1.18), *fas* (−1.17), *rac1l* (−1.13), *fgf6a* (−1.09), and *cdc42l2* (−1.08) were decreased, while *syngr2b* (1.04), *dusp1* (1.29), *nr4a1* (1.58), *pla2g4f-like* (2.16), and *fosab* (2.48) were increased, suggesting that loss of GCGR affected MAPK signaling pathway (Figure 4C).

To experimentally validate the dysregulation of lipid metabolism in *gcgr^−/−^* zebrafish, we performed oil-red O staining in whole mount and frozen sections in the wildtype and *gcgr^−/−^* fish in 7 dpf. The results showed a significant accentuation of fat in the liver (Figure 4D–G). However, the fat accumulation in other tissues were not significantly different (Figure 4D–G). These results suggest that GCGR is mainly required for lipid metabolism in the liver.

### 2.5. GCGR Regulates Carbohydrate Metabolism

Thirty DEGs in 14 carbohydrate metabolic pathways were enriched (Figure 5A). They include glycolysis/gluconeogenesis (ko00010), citrate cycle (TCA cycle) (ko00020), pentose and glucoronate interconversions (ko00040), fructose and mannose metabolism (ko00051), galactose metabolism (ko00052), ascorbate and aldarate metabolism (ko00053), starch and sucrose metabolism (ko00500), amino sugar and nucleotide sugar metabolism (ko00520), pyruvate metabolism (ko00620), glyoxylate and dicarboxylate metabolism (ko00630), propanoate metabolism (ko00640), butanoate metabolism (ko00650), as well as C5-branched dibasic acid metabolism (ko00660) (Appendix A).

Four DEGs in glycolysis/gluconeogenesis pathway were enriched, two upregulation and two downregulation. The upregulated genes were *adh5* (alcohol dehydrogenase 5) (2.01), which converts alcohol to aldehydes or ketones, and *acss2l* (acyl-CoA synthetase short chain family member 2 like) (1.37), which produces acetyl-CoA from acetate during gluconeogenesis. The two downregulated genes were *adpgk2* (ADP-dependent glucokinase 2) (−1.41), which catalyzes ADP and D-glucose to AMP and D-glucose 6-phosphate during glycolysis, and *aldh2.2* (aldehyde dehydrogenase 2 family member, tandem duplicate 2) (−1.11) which is responsible for the conversion of acetaldehyde to acetate and participate in glycolysis/gluconeogenesis pathways. 

DEGs were also enriched in other pathways of carbohydrate metabolism. In Pyruvate metabolism pathway (ko00620), *me1-like* (malic enzyme 1 like), which catalyzes malate to pyruvate, was markedly increased (6.57). Interestingly, one gene in the glycogenesis, *gyg2* (glycogenin 2) (1.69) was also upregulated. Moreover, ten UDP-Glucuronosyltransferases in the pentose and glucoronate interconversions pathway (ko00040) and ascorbate and aldarate metabolism pathway (ko00053) (*ugt1a1*, *ugt1a2*, *ugt1ab*, *ugt1a7*, *ugt1b2*, *ugt2a2*, *ugt2a4*, *ugt2b3*, *ugt5b1*, *ugt5b4*) were all dramatically decreased. These UGTs are mainly expressed in liver, which are phase II biotransformation enzymes that regulate the glucuronidation reaction [29]. Taken together, these data suggested that GCGR is essential to maintain carbohydrate metabolism and its deficiency results in the disorder of carbohydrate metabolism.

The *gcgr^−/−^* zebrafish has lower total free glucose than the wildtype control [25]. Although increased GLP-1 is likely the major culprit [30,31], we also investigated whether increased glucose uptake in *gcgr^−/−^* zebrafish may contribute to the hypoglycemia, as well. Therefore, we performed a 2-NBDG uptake assay using wildtype and *gcgr^−/−^* larvae. The result showed that the *gcgr^−/−^* zebrafish had increased glucose uptake compared with WT, as indicated by the fluorescence intensity of lens [32] (Figure 5B,C). Additionally, when cultured in a medium with high glucose (20 mM), the α-cell hyperplasia was partially suppressed (Figure 5D–F). These results suggest that hypoglycemia may be permissive or necessary for α-cell hyperplasia. 

### 2.6. GCGR Regulates Amino Acid Metabolism

KEGG enrichment analysis identified two categories related to the metabolism of standard amino acids and other amino acids (Figure 6, Appendix A). There were 21 downregulated and five upregulated DEGs related to metabolism of the 20 standard amino acids. The top five downregulated genes are *med15-like* (−4.96), *ercc4-like* (−4.72), *il4i1*(−4.68), *nags* (−2.03), and *kyat3* (−1.98) (Figure 6). Gene *il4i1*, defined as L-amino-acid oxidase (LAAO), was severely downregulated. The enzyme is involved in the regulation of many amino acid pathways, including alanine, aspartate, and glutamate metabolism pathway (ko00250), cysteine and methionine metabolism pathway (ko00270), valine, leucine and isoleucine degradation pathway (ko00280), tyrosine metabolism pathway (ko00350), phenylalanine metabolism pathway (ko00360), tryptophan metabolism pathway (ko00380), phenylalanine, as well as tyrosine and tryptophan biosynthesis pathway (ko00400) (Appendix A and Figure 6A). Moreover, *cthl* (cystathionine gamma-lyase-like), which encodes an enzyme that breaks down cystathionine into cysteine, α-ketobutyrate, and ammonia, was also downregulated. In addition, *ido1* (indoleamine-pyrrole 2,3-dioxygenase), which regulates the O2-dependent oxidation of L-tryptophan to N-formylkynurenine, was decreased (−1.16). The five upregulated genes included *cspg4-like* (1.65), *adh5* (2.01), *setd8-like* (2.43), *dicp1.19* (4.91), and *mll3-like* (6.76) (Figure 6A). 

Expression of genes related to metabolism of other amino acids was also altered in *gcgr^−/−^* mutant (Appendix A and Figure 6B). For glutathione metabolism (ko00480), *ercc4-like* (−4.72), *rrm2* (−2.92), *gstt2* (−1.79), *gpx2* (−1.26), *gstm.1* (−1.20), *rrm1* (−1.19), *ggt1a* (1.10), *cspg4-like* (1.65) were affected. Two of the glutathione S-transferase (*gstt2* and *gstm.1*), which catalyze conjugation of reduced glutathione to xenobiotic substrates for the purpose of detoxification [33], were significantly decreased. While *ggt1a* (gamma-glutamyltransferase 1), which hydrolyzes the γ-glutamyl bond of glutathione to produce glutamate, cysteine (cystine), and glycine [34], was increased. All these data suggested the disorder of amino acid metabolism in *gcgr^−/−^* mutant. 

It is known that the blood free amino acid levels are increased in the GCGR knockout mice. This is concomitant with a decrease of intracellular amino acid content in hepatocytes [35]. However, it is unclear whether there is a decrease in the whole-body amino acid content. Here, we measured the total amino acid compositions of the whole organism, and found that Met, Ile, Tyr were slightly decreased in the *gcgr^−/−^* mutant zebrafish, while other amino acids did not significantly change (Figure 6C).

### 2.7. Verification of Transcriptome Data by qRT-PCR

To further evaluate our RNA-seq data, we performed qRT-PCR analysis of selected genes in categories of lipid metabolism (*lipca*, *elovl7b*, *apobb.2*, *fasn-like*, *hadhaa*, and *pcsk9*), carbohydrate metabolism (*adh5*, *acss2l*, *fuk*, *gale*, and *mlycd*), amino acid metabolism (*cspg4-like*, *il4i1*, *papss1*, *cthl*, *dnmt1*, and *eevs*), and five other genes (*gcgb*, *cox11*, *pvalb3*, *slc2a1a*, and *ndufs8b*). Our qRT-PCR results were consistent with the RNA-seq data (Figure 7A–D). The linear regression analysis between RNA-seq and qRT-PCR data was statistical significance (*r* = 0.9591, *p* < 0.0001) (Figure 7E).

## 3. Discussion

Glucagon is a counter-regulatory hormone in glucose homeostasis. It primarily acts on liver GCGR to increase blood glucose by promoting glycogenolysis and gluconeogenesis [1]. Therefore, most studies focus on the GCGR function in the liver. However, it should be appreciated that GCGR is also detectable in many other tissues, including adipose tissue, brain, kidney, heart, and gastrointestinal tract [25,36]. Metabolism is a complex process regulated by the crosstalk of multiple tissues, a systemic dissection of the metabolic network may help better understand the physiological function of GCGR. In this study, we performed RNA-seq analysis using the whole organism of GCGR knockout zebrafish, and found that it regulates the lipid, carbohydrate, and amino acid metabolism networks.

The lipid metabolism was disrupted in the *gcgr^−/−^* knockout zebrafish. Fifty-two genes related to lipid metabolism were altered more than two-fold. Except for an increase of *elovl7b* in fatty acid elongation pathway, 11 genes enriched in fatty acid related metabolism were all decreased (Figure 4A). That data suggest that fatty acid anabolism and catabolism are all decreased in the *gcgr^−/−^* mutant. For fatty acid oxidation, the *hadhaa* (hydroxyacyl-CoA dehydrogenase trifunctional multienzyme complex subunit alpha a), whose product catalyzes the last three steps of mitochondrial β-oxidation of long chain fatty acids, was significantly decreased. In addition, the *acox1*(peroxisomal acyl-coenzyme A oxidase 1), which also plays a crucial role in the fatty acid β-oxidation, was also downregulated (Figure 4A). Taken together, these data suggest that the *gcgr^−/−^* mutants have decreased fatty acid β-oxidation compared to the wildtype fish. 

We further found that the *gcgr^−/−^* mutants display a significant accumulation of lipids in the liver (Figure 4D–G). Why lipids accumulate in the *gcgr^−/−^* liver is unknown. One possibility is that the *gcgr^−/−^* mutant may increase cholesterol metabolism based on our data. The dramatic increase of *ldlr* mRNA level in the *gcgr^−/−^* zebrafish predicts an elevated absorption of LDL-cholesterol (LDL-C) into hepatic cells. The decrease of *pcsk9*, which degrades LDLR, may further increase hepatic cholesterol. Consistent with our results, the GCGR antagonist-treated mice induces increased liver cholesterol absorption [17]. Additionally, T2D patients treated with glucagon receptor antagonist LY2409021 resulted in a statistically significant increase in hepatic fat fraction in a clinic trial [37]. Taken together, these data suggested that disruption of GCGR causes aberrant expression of lipid metabolism genes, which increased cholesterol absorption and resulted in the accumulation of lipid in the hepatic cells.

Glucose level is lower in GCGR knockout mouse, as well as zebrafish [6,25]. Additionally, the *gcgr^−/−^* mice are resistant to develop hyperglycemia in STZ-induced T1D and high-fat diet induced T2D models [7,8]. This phenotype has been attributed the associated increase of plasma GLP-1 levels [6,7,31,38]. Moreover, recent studies revealed that the improvement of glucose control by GCGR blockade required remnant insulin action in the diabetic animals [39,40]. Overall, previous studies have revealed that GCGR plays an important role in glucose metabolism. In our studies, the mRNA levels of most of the key enzymes in the glycolysis and gluconeogenesis pathways were unchanged, such as *gck* (−0.006), *g6pca.1*(0.38), *g6pca.2* (0.31), *g6pcb* (0.30), *gpia* (0.07), *pfkla* (−0.17), *pfklb* (−0.09), *aldoaa* (0.06), *aldoab* (−0.01), and *pgk1* (−0.05). However, the mRNA levels of several regulators of these enzymes were enriched more than two-fold in the glycolysis/gluconeogenesis pathway. The *gckr* (glucokinase regulator), which forms an inactive complex with glucokinase, were downregulated (−1.27), suggesting improved GCK activity in *gcgr^−/−^* animals. By contrast, the *adpgk2* (ADP-dependent glucokinase 2) was downregulated (−1.41). Moreover, three other genes, *aldh2.2*, *adh5*, and *acss2l* were also affected. However, limited gene information prevents us from making a conclusive statement on the global glycolysis/gluconeogenesis change in the *gcgr^−/−^* mutants. Further information, such as global proteomic data can help us answer the question. In addition to glycolysis/gluconeogenesis pathway, 26 genes in other pathways of carbohydrate metabolism were also altered, suggesting that GCGR signaling may affect these genes, directly or indirectly, and they may contribute to the low blood glucose and the increased glucose tolerance. 

We found that the *gcgr^−/−^* mutant zebrafish have an increased uptake of exogenous glucose using 2-NBDG as a surrogate. Similar results were found in mice. The GCGR antagonist REMD2.59 improves whole-body insulin sensitivity and glucose uptake in the *ob/ob* T2D model [41]. Moreover, when we cultured the *gcgr^−/−^* mutant larvae in a high glucose medium for three days, the α-cell hyperplasia was suppressed in the mutant (Figure 5F). These data suggest that low glucose levels facilitate α-cell hyperplasia. 

Several studies have confirmed that amino acid catabolism is decreased in mice due to decreased expression of transporters in the liver when GCGR signaling is disrupted, leading to increased plasma free amino acid levels [42,43,44,45]. Glutamine and alanine are dramatically increased and both contribute to α-cell hyperplasia [42,43]. In our study, the transcription level of several key amino acid catabolism genes was significantly decreased. For instance, L-amino acid oxidase gene *il4i1*, which catalyzes the oxidative deamination of a number of L-amino acids, was 26.4-fold decreased. Another gene, *ido1* that regulates the rate-limiting step of oxidative tryptophan degradation, decreased 2.3-fold. Moreover, other amino acid metabolism genes were also dramatically affected. There was a 2.3-fold decrease in *alas2* (Delta-aminolevulinate synthase 2), which catalyzes the condensation of glycine and succinyl coenzyme A to form 5-aminolevulinic acid (ALA) during heme biosynthesis. The gene *gatm* (guanidinoacetate N-methyltransferase), which catalyzes creatine synthesis from amino acids glycine, arginine, and methionine, was downregulated 2.4-fold. Moreover, *nags* (n-acetylglutamate synthase) that produces N-acetyl glutamate, which regulates the key enzyme of ureagenesis CPS1 (carbomyl phosphate synthetase-1) during ureagenesis, was decreased 4.1-fold. However, the total amino acid content only decreased slightly in *gcgr^−/−^* mutants. This reflects the importance of intracellular amino acid homeostasis. Nevertheless, all these data suggested that the zebrafish amino acid catabolism and related metabolism are downregulated. These changes most likely stem from the liver, similar to the mouse.

In summary, we performed whole-organism RNA-seq of wildtype and *gcgr^−/−^* mutant larvae. Compared with wildtype, the mutant data showed that many genes in metabolic processes were changed more than two-fold. By further analysis, we found that most of these genes are related to lipid metabolism, carbohydrate metabolism, and amino acid metabolism. Of note, genes related to fatty acid β-oxidation, amino acid catabolism, and ureagenesis were downregulated. Moreover, we experimentally confirmed that the mutants display an increased accumulation of lipids in the liver, an increase in glucose uptake, and modest changes in total amino acids. These results provide new information about the global metabolic network that GCGR signaling regulates in addition to a better understanding of the receptor’s physiological functions. Nevertheless, we still do not know that these changes in the *gcgr^−/−^* mutants were due to the direct or indirect effects of GCGR deficiency. Further studies need to perform transcriptomic analyses of the *gcgr^−/−^* mutants at different stages, as well as in different tissues.

## 4. Materials and Methods

### 4.1. Zebrafish Lines and Maintenance

Zebrafish (Danio rerio) were raised in a recirculating aquaculture system (Shanghai Haisheng Biotech Co., Ltd., Shanghai, China) on a 14:10 h darkness cycle at 28 °C. Embryos were obtained from natural breeding and raised at 28.5 °C in an embryo rearing solution, and staged according to Kimmel et al. (1995) [46]. In this study, the AB strain, *gcgra^−/−^*; *gcgrb^−/−^* double mutant fish (referred as *gcgr^−/−^* henceforth) [25], and *Tg(gcga:GFP)* [47] were used. All procedures have been approved by the Xiamen University Institutional Animal Care and Use Committee (Protocol XMULAC20160089, 10 March 2016).

### 4.2. RNA Extraction, cDNA Library Preparation, Sequencing

Total RNA was isolated from pools of 30 larvae using TRIzol reagent (Thermo Fisher Scientific, Waltham, MA, USA) according to the manufacturer’s instruction. Three biological repeats were performed for each genotype. To remove any genomic DNA contamination, the RNA were then digested using the RQ1 RNase–Free DNase (Promega, Madison, WI, USA). The concentration and quality of each sample were determined by an Agilent RNA 6000 nano kit in the Agilent 2100 bioanalyzer (Agilent Technologies, Santa Clara, CA, USA). The RNA quality score (RIN/RQN) was 10 for all of the samples.

To construct cDNA libraries, the mRNA of each sample was selected using Oligo (dT) magnetic beads. The total mRNA was then fragmented and reverse-transcribed into double-strand cDNA (ds cDNA) by N6 random primers. The synthesized double-strand cDNA was subjected to end repair, follow by 5′ phosphorylation and 3′ adenylation. Next, an adapter was ligated to the 3′ adenylated cDNA fragments. The ligation products were used for PCR amplification for enrichment of cDNA templates. Following denaturation of these PCR products, the single stranded DNA was then circlized to form the final cDNA library. All the sequencing of these cDNA libraries was performed on a BGISEQ-500 RS platform (BGI, Shenzhen, China) according to the manufacturer’s stand protocol [48]. 

### 4.3. Bioinformatics Analysis of RNA Sequence Data

Raw reads were subjected to a BGISEQ-500 quality control test and filtered into a clean read using SOAPnuke software [49]. This process discarded reads containing adaptor sequences, reads containing unknown bases “N” more than 10%, and low quality reads (the percentage of low quality bases more than 50% in a read). After filtering, the clean reads were aligned with the zebrafish genome (Danio rerio, GRCz11, http://asia.ensembl.org/Danio_rerio/Info/Index) using HISAT (v2.0.4) [26]. Following alignment, the clean reads to zebrafish unigenes using Bowtie2 [27], RSEM (RNA-Seq by expectation maximization) [50] was performed to quantify the gene expression using the FPKM method (fragments per kilobase of transcript per million fragments sequenced) [51]. Moreover, some uncharacterized genes were annotated in the Ensembl (www.ensembl.org/) by BLAST and synteny analysis. These sequence data were deposited in an NCBI database and are accessible via BioProject ID: PRJNA541367.

### 4.4. Analysis of Differentially Expressed Genes (DEGs)

The DEseq2 was used to identify DEGs [52]. Genes with fold change ≥ 2.00 and adjusted *p*-value ≤0.05 were considered as differentially expressed genes (DEGs) with statistical significance. For duplicate samples, the log2 fold change (Log2FC) and probability for each gene in every comparison was calculated by the NOIseq under the conditions with fold change ≥ 2.00 and probability ≥ 0.8 [53].

To perform the pathway enrichment analysis, the significant DEGs were subjected to GO (gene ontology) and KEGG (Kyoto encyclopedia of genes and genomes) database analysis. The analysis was performed using the hypergeometric test and false discovery rate (FDR) correction methods. Significantly enriched genes were those with a *p*-values < 0.05 and a FDR ≤0.01. 

### 4.5. Quantitative RT-PCR

To validate the DEGs, qRT-PCR was performed. To synthesize the first strand cDNA, the RNA was reverse transcribed by the M-MLV (Promega, Madison, WI, USA) with oligo(dT)16 primers. The qRT-PCR reaction was carried out in an Agilent AriaMx system (Agilent Technologies, Santa Clara, CA, USA) using powerUp SYBR Green Master Mix (Thermo Fisher Scientific, Carlsbad, CA, USA). The amplification program was 95 °C, 10 s; 60 °C, 30 s for 40 cycles. Three biological samples of wildtype or *gcgr^−/−^* were used, and each assay for a sample was performed in triplicate. mRNA levels were calculated using the 2^−ΔΔ*C*t^ method [54] and presented as relative (fold) levels normalized to the level of β-actin. The data were presented as the mean of three biological samples. The primers used were listed in Appendix A.

### 4.6. Oil Red Staining

Zebrafish at 7 dpf were fixed in 4% paraformaldehyde in PBS for 1 h at room temperature. The fish were washed with PBS subsequently to remove paraformaldehyde, incubated in 60% 2-propanol for 10 min, followed by freshly filtered 0.3% oil red O (ORO) (Xiya Reagent, Chengdu, China) in 60% 2-propanol for 10 min. The staining was terminated by washing twice with 60% 2-propanol. The samples were transferred to 75% glycerol and imaged using Lecia M205 FCA (Lecia Wetzlar, Germany).

### 4.7. 2-NBDG Uptake Test

Zebrafish (5 dpf) were incubated in a culture medium containing 600 uM 2-NBDG (Apexbio, B6035, Houston, TX, USA) for 3 h. The larvae were anesthetized for imaging under a M205 FCA microscope (Leica, Wetzlar, Germany), the fluorescence intensity of lens was used as the indicator of glucose uptake according to the reference [32].

### 4.8. Glucose Exposure and α-Cells Counting

The *Tg(gcga:GFP)* and *gcgr^−/−^*; *Tg(gcga:GFP)* larvae were incubated with 20 mM glucose or a 0.3× Danieau solution from 4 to 7 dpf for three days. After harvest, larvae were fixed with 4% paraformaldehyde overnight at 4 °C, and then placed on a slide with aqua-mount (Richard-Allan Scientific, Kalamazoo, MI, USA) with the right side of larvae up to expose the islet. The α-cell number was counted according to the GFP under a Zeiss AxioImager A1 microscope (Carl Zeiss, Jena, Germany) with 40× lens.

### 4.9. Behavior Test

Seven dpf zebrafish were transferred individually to a compartment containing 1 mL 0.3X Danieau solution in a 24-well plate. Then, the plate was video-recorded by a top-view camera in the DanioVision tracking system (Noldus IT, Wageningen, Netherlands) for 3 min, the moving tracks were generated and the total moving distance and velocity were determined by Ethovision XT7 software (Noldus IT, Wageningen, Netherlands), 48 animals from each group were analyzed.

### 4.10. Amino Acid Measurement

Seventy embryos of 7 dpf for each group were harvested in 1.5 mL Eppendorf tubes. After lyophilization, samples were hydrolysed in 6 N HCl at 110 °C for 24 h. Then, the volume was adjusted to 10 mL by adding distilled water. One ml of hydrolysate was dried using nitrogen blower and resuspended in 1 mL 0.02 M HCl. After resuspension, samples were filtered through 0.22 μm filters, the amino acid compositions were determined by an automatic amino acid analyzer (L-8900, HITACHI, Chiyoda, Japan). For WT and *gcgr^−/−^*, three independent biological repeats were performed. Amino acid in the embryos were expressed as µg/mg wet tissue.

### 4.11. Statistical Analysis

Data were presented as means or means ± S.E.M. The statistics were performed using one-way ANOVA followed by Newman–Keuls post-hoc test or *t*-test (SPSS). *p* < 0.05 was considered as significant.

## Figures and Tables

**Figure 1 ijms-21-00724-f001:**
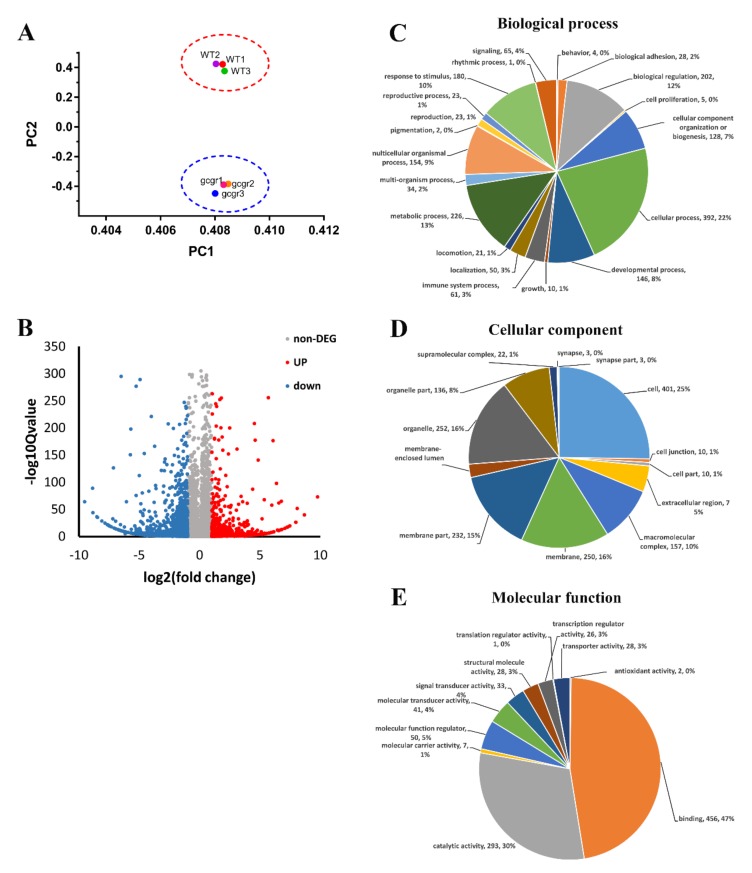
RNA-seq (RNA sequencing) analysis of glucagon receptor (*gcgr^−/−^*) mutant zebrafish. (**A**) Principal component analysis (PCA) plot of three wildtype and *gcgr^−/−^* mutant RNA-seq datasets. Principal component 1 (PC1), and principal component 2 (PC2) were used for analysis. (**B**) Volcano plot of differential expression analysis of *gcgr^−/−^* mutant and control larvae showing the relationship between *p*-value and log fold changes. Red shows upregulated genes and blue downregulated genes. (**C–E**) Gene ontology (GO) enrichment analysis of differentially expressed genes (DEGs) in *gcgr^−/−^* mutant zebrafish. The DEGs were assigned to three categories: Biological process, (**C**) cellular component, (**D**) and molecular function (**E**). The names of the GO subcategories, the number of genes, and the proportion of each subcategory are listed by the pie charts.

**Figure 2 ijms-21-00724-f002:**
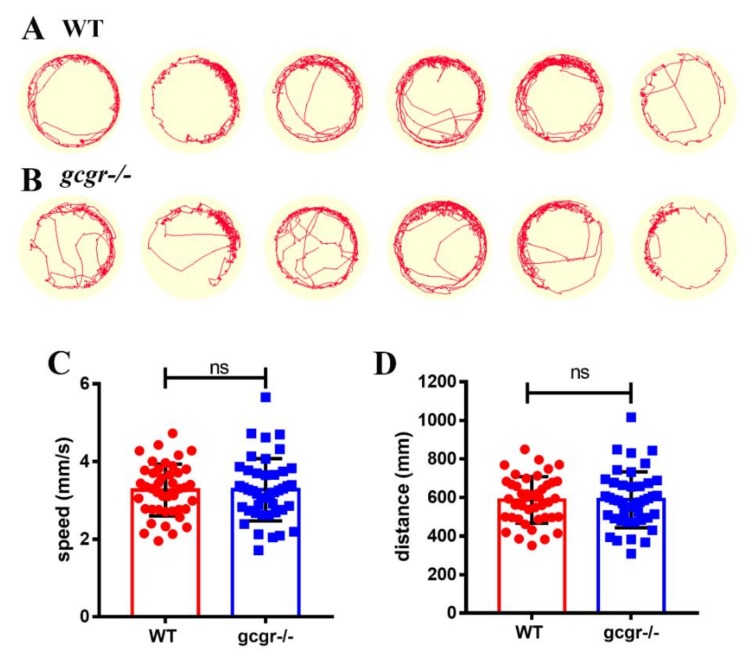
*gcgr^−/−^* mutant fish have similar physical activity to controls. (**A**,**B**) Representative images of activity tracks of six wildtype (**A**) and *gcgr^−/−^* (**B**) zebrafish larvae. Each image is the track of one larva recorded for 3 min. (**C**,**D**) The average swimming speed (**C**) and swimming distance (**D**) of wildtype and *gcgr^−/−^* zebrafish larvae. Ns: no significance.

**Figure 3 ijms-21-00724-f003:**
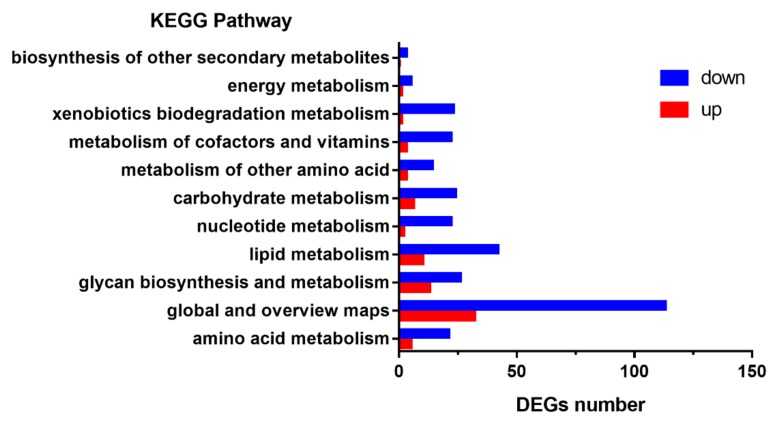
Kyoto encyclopedia of genes and genomes (KEGG) enrichment analysis of DEGs in metabolism pathways. The y-axis indicates pathways and the x-axis indicates the number of DEGs. The red bar shows the upregulated genes and the blue bar shows the downregulated genes.

**Figure 4 ijms-21-00724-f004:**
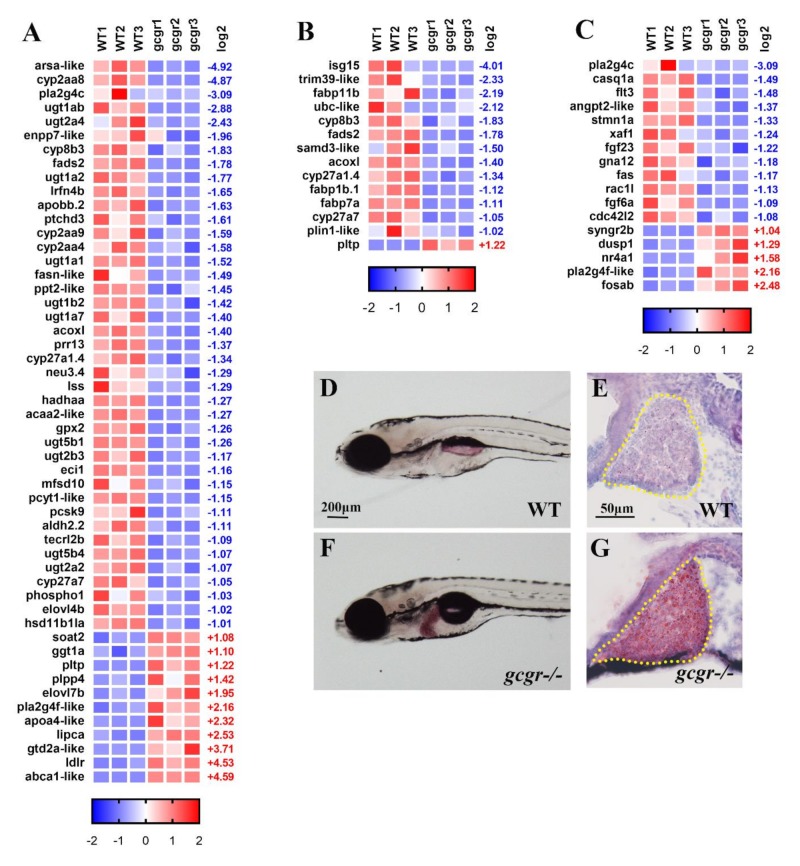
GCGR regulate lipid metabolism in zebrafish larva. (**A–C**) Heatmaps of transcripts in lipid metabolism enrichment (**A**) in peroxisome proliferator-activated receptors (PPAR) pathway enrichment (**B**) and in mitogen-activated protein kinase (MAPK) pathway enrichment (**C**). Colors represent high (red), low (blue), or average (white) expression values based on Z-score normalized fragments per kilo base per million mapped reads (FPKM) values for each gene. The Z-score indicators are shown under each map. The fold change (log2) are shown on the right. (**D–G**) Oil Red O (ORO) staining of 7 dpf wildtype (**F**) and *gcgr^−/−^* (**G**) mutant larvae. Whole-mount ORO staining (**D**,**F**) and liver frozen sections ORO staining (**E**,**G**) both show accumulation of lipids in the mutant liver. Yellow dash line indicates the liver area.

**Figure 5 ijms-21-00724-f005:**
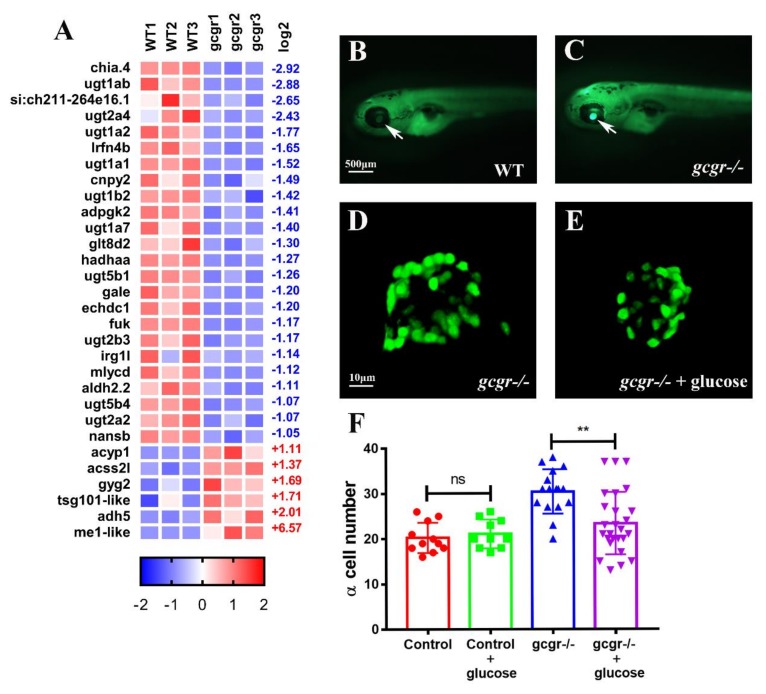
GCGR regulates carbohydrate metabolism in zebrafish larvae. (**A**) Heatmaps of transcripts in carbohydrate metabolism enrichment. Colors represent high (red), low (blue), or average (white) expression values based on the Z-score normalized FPKM values for each gene. The Z-score indicators are shown under each map. The fold change (log2) are shown on the right. (**B,C**) 2-NBDG glucose uptake of wildtype (**B**) and *gcgr^−/−^* mutant (**C**) larvae, the glucose uptake level is indicated by the fluorescence of lens (arrow) imaged by fluorescent microscopy. (**D–E**) Representative images of the principal islet of *gcgr^−/−^*; *Tg(gcga:GFP)* (**D**) and 20 mM glucose treated-*gcgr^−/−^*; *Tg(gcga:GFP)* (**E**). The images are confocal projections; scale bars indicate 10 μm. (**F**) Quantification of the α-cell number in different groups of zebrafish at 7 dpf, *n* ≥ 10. Ns: No significance, and ** *p* < 0.01 by *t*-test.

**Figure 6 ijms-21-00724-f006:**
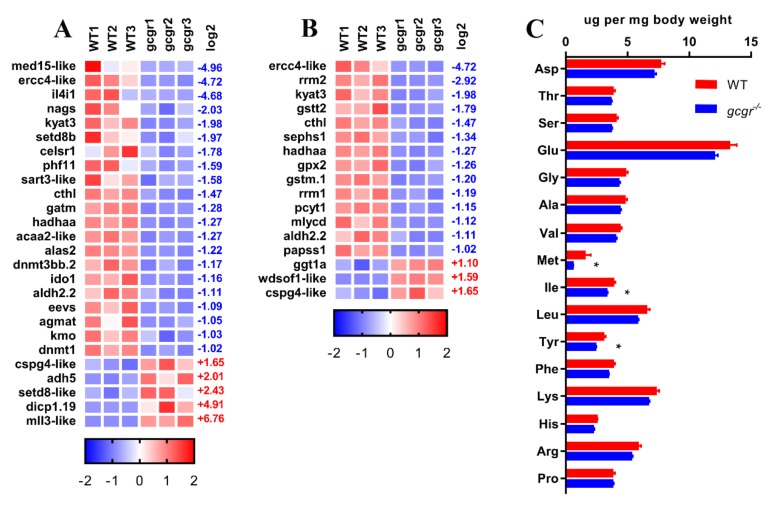
GCGR regulates amino acid metabolism in zebrafish larvae. (**A**) Heatmaps of transcripts in amino acid metabolism enrichment. (**B**) Heatmaps of transcripts in metabolism of other amino acid enrichment. (**C**) Amino acids compositions in WT and *gcgr^−/−^* embryos. Results were represented as means with standard errors (*n* = 3), * *p* < 0.05 by *t*-test.

**Figure 7 ijms-21-00724-f007:**
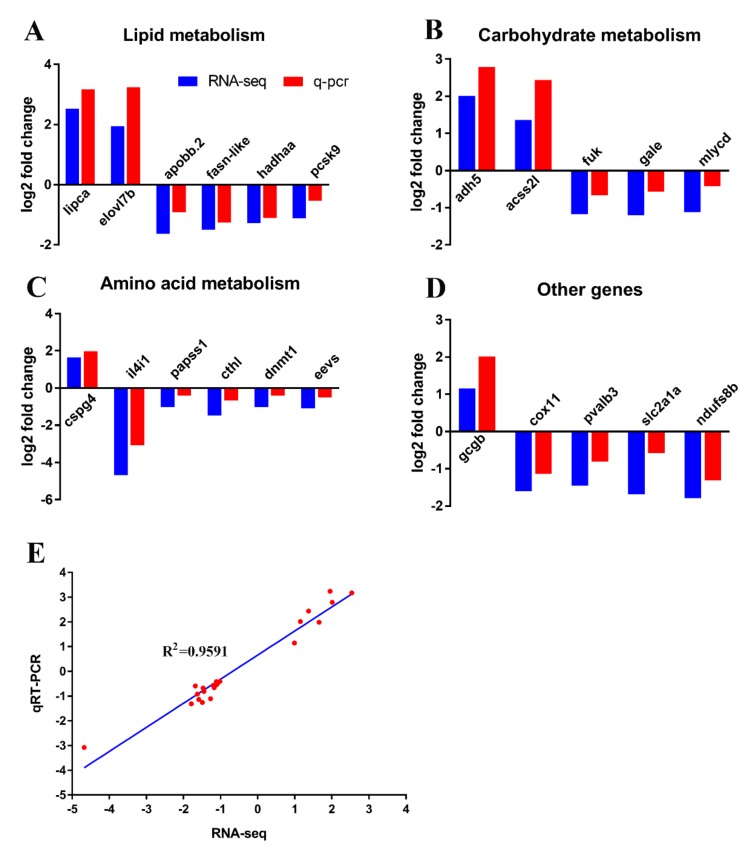
Validation of RNA-seq data using qPCR. (**A–D**) The validation of the expression levels of differentially expressed genes by qRT-PCR analysis in the categories of lipid metabolism (**A**) carbohydrate metabolism, (**B**) amino acid metabolism, (**C**) and other DEGs (**D**). (**E**) Correlation between qRT-PCR and RNA-seq results for select DEGs. The log2 (fold change) values derived from the RNA-seq analysis of DEGs are compared with those obtained by qRT-PCR determined by 2^−ΔΔ*C*T^. The reference line indicates the expected linear relationship.

**Table 1 ijms-21-00724-t001:** Statistics for read filtering and mapping.

Sample Name	WT1	WT2	WT3	gcgr1	gcgr2	gcgr3
Total raw reads (M)	21.94	21.94	21.94	21.94	21.94	21.94
Total clean reads (M)	21.89	21.9	21.92	21.93	21.89	21.89
Clean reads ratio (%)	99.76	99.82	99.9	99.92	99.75	99.74
Genome total mapping (%)	90.38	89.55	90.25	90.54	90.47	90.55
Genome uniquely mapping (%)	64.66	63.98	64.69	65.12	65.53	65.34
Genes total mapping (%)	79.41	79.83	79.28	80.03	79.62	79.89
Genes uniquely mapping (%)	69	69.02	69.03	69.21	69.13	69.21

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
