# Peer review of "Global Transcriptomic Analysis of Zebrafish Glucagon Receptor Mutant Reveals Its Regulated Metabolic Network"

_ijms, 2020, doi:10.3390/ijms21030724_

Round 1

Reviewer 1 Report

This manuscript by Kang et al., addresses and interesting question in developmental biology and metabolism: the role of glucagon transporter in metabolism. However, several key points must be addressed prior publication in IJMS.

Major points:

The English of the manuscript must be improved. It has to be corrected professionally or by a native English speaker It is unclear why the author decided to perform all their analysis at 7dpf. Why so late? What is the rational for this? How do they know that they are not detecting secondary effects of glu receptor depletion? When do the gcgr mutants die? Were the larvae fed or not prior RNA extraction? If no, why did the authors decide to fast the larvae, if Yes what type of food and when were they fed last? Figure 1 legend: a 7 dpf zebrafish is a larvae not an embryo Figure 1 legend: Define GO Figure 2: was this analysis performed at 7 dpf? P4: define KEGG Figure 4D: why is the yolk sac of the control larvae so big compared to F? Moreover the craniofacial stage of the control in D looks like a much younger larvae than 7 dpf Figure 4E: Is this the liver of a 7 dpf? Figure 4: why do the FGF21 mechanisms mention later are not addressed in the Heat maps? How long was the glucose exposure in Fig 5? Section 2.7: why did the author select those genes? Figure 7: this figure has no stats nor error bars. How was Figure 7E generated without any statistical analysis in Figure 5 A-D? P11: Can the author speculate why gcgr mutants have lipids accumulation in the liver? Section 4.5: no statistical analysis in mentioned for the qPCR analysis Section 4.7: why was his analysis performed at 5 dpf and not 7 dpf like the rest of the analysis?

Minor points:

Please decide between α cell and α-cell P2 define dpf

Author Response

The English of the manuscript must be improved. It has to be corrected professionally or by a native English speaker.

A native English speaker has carefully revised our manuscript according to your suggestion.

It is unclear why the author decided to perform all their analysis at 7dpf. Why so late? What is the rational for this?

We thank the reviewer for the questions. Blood glucagon is highest when the animal is fasting. Therefore, glucagon signaling is likely most active at this stage. At 7 dpf, the larva has exhausted the yolk and likely in a fasting state. Indeed, our previous study (Li Mingyu et al. Journal of Endocrinology, 2015, 227, 93-103), showed that 7 dpf appears to be an ideal time point for the study of α-cell hyperplasia phenotypes. That is why we use the 7 dpf larvae for the transcriptiome analysis.

How do they know that they are not detecting secondary effects of glu receptor depletion?

We do not know that the changes in the gcgr mutant were due to the primary or secondary effects. We added a sentence in the revised manuscript’s “Discussion” section to highlight this possibility.

When do the gcgr mutants die?

The gcgr mutants develop normally and can survive to adulthood without any obvious developmental defects.

Were the larvae fed or not prior RNA extraction? If no, why did the authors decide to fast the larvae, if Yes what type of food and when were they fed last?

The larvae were not fed prior the RNA extraction. We did not feed them to maintain them in fasting state. Why we chose this stage for the RNA-seq was answered in the question 2.

Figure 1 legend: a 7 dpf zebrafish is a larvae not an embryo.

Thanks. We made change according to your suggestion.

Figure 1 legend: Define GO.

Thanks. We made change according to your suggestion.

Figure 2: was this analysis performed at 7 dpf?

Yes, it was.

P4: define KEGG

Corrected

Figure 4D: why is the yolk sac of the control larvae so big compared to F? Moreover the craniofacial stage of the control in D looks like a much younger larvae than 7 dpf

We appreciate this criticism for the carefully observation. According to the reviewer’s criticism, we re-performed the oil red staining and used new images in the new Figure 4D and 4F, please see the revised Figure 4 in the revision for detail.

Figure 4E: Is this the liver of a 7 dpf?

Yes, it is.

Figure 4: why do the FGF21 mechanisms mention later are not addressed in the Heat maps?

We thank this reviewer for raising this point. Since the KEGG did not find the FGF21pathway enriched in our results, we therefore did not address in our heatmaps.

How long was the glucose exposure in Fig 5?

Sorry for the missing information. The larvae were exposed in glucose from 4dpf to 7dpf for 3 days. We added a new section to describe the procedure of this experiment. Please see section 4.8 in the revision.

Section 2.7: why did the author select those genes?

When we selected genes for qPCR, we chose ones that are important in the pathways/ processes that we are interested in.

Figure 7: this figure has no stats nor error bars. How was Figure 7E generated without any statistical analysis in Figure 7 A-D?

Actually, all the RNA-seq and qPCR data were obtained from three biological repeats. For the RNA-seq, each gene in a sample had the single value, and the data of specific gene were presented as the log2 value which resulted from the mean of three gcgr mutant samples compared to mean of three wild type samples. So, there is only one log2 value from the RNA-seq. Although we were able to put the error bars for the data of the qRT-PCR, we presented both without error bars to keep it consistent. To our knowledge, most of publications present similar data in this way. For Figure 7E, we added the R2 for further analysis.

P11: Can the author speculate why gcgr mutants have lipids accumulation in the liver?

One possibility is that the gcgr mutant may increase cholesterol metabolism based on our data. The dramatic increase of ldlr mRNA level in the gcgr-/- zebrafish predicts an elevated absorption of LDL-cholesterol (LDL-C) into hepatic cells. The decrease of pcsk9, which degrades LDLR, may further increase hepatic cholesterol. Consistent with our results, the GCGR antagonist treated mice induces increased liver cholesterol absorption (Guan et al., 2015). Additionally, T2D patients treated with glucagon receptor antagonist LY2409021 resulted in a statistically significant increase in hepatic fat fraction in a clinic trial (Guzman et al., 2017). Taken together, these data suggested that disruption of gcgr causes aberrant expression of lipid metabolism genes, which increased cholesterol absorption and resulted in the accumulation of lipid in the hepatic cells.

We have added this paragraph in the Discussion part in the revision.

   17. Section 4.5: no statistical analysis in mentioned for the qPCR analysis

Sorry for missing that. Three biological samples of wild type or gcgr-/- were performed, and each assay for a sample was performed in triplicate. The data are presented as the mean of three biological samples in Figure 7 A-D. We added these information in the revision.

   18. Section 4.7: why was his analysis performed at 5 dpf and not 7 dpf like the rest of the analysis?

Thanks for this criticism. We initially performed this experiment at 3 dpf according to the literature (Lee J et al. ACS Chem. Biol. 2013, 8, 8, 1803-1814.). However, we did not see a difference between wild type and gcgr mutant, probably due to minimal glucagon function as there are plenty of yolk supply and the fish is at fed state. We therefore performed the experiment at 5dpf and found an obviously difference.  7 dpf may be a better time point.

Minor points:

1.Please decide between α cell and α-cell

Thanks for raising this point. Throughout the manuscript, we use “α cell” or “α cells” as a noun and "α-cell" as an attributive noun. We have revised these typos.

P2 define dpf

Corrected 

Reviewer 2 Report

The authors present RNASeq data from a previously described mutant in the 

glucagon receptor (GCGR) gene. They focus on the analysis and differential expression of several metabolic genes and pathways. 

The data presented is very interesting, however the authors do not discuss in detail why they chose this particular stage (7days) to study the transcriptomics profile of the mutants.

Although there is no gross morphology phenotype, the liver shows already signs of steatosis and some of the changes in the transcriptomics profile they observe might be secondary effects and not the primary consequence of a GCGR knock out. 

This should be further discussed and explicitely mentioned as a limitation of the current study (that would require different - earlier- timepoints of transcriptomic analyses to address)

Author Response

The data presented is very interesting, however the authors do not discuss in detail why they chose this particular stage (7days) to study the transcriptomics profile of the mutants.

We thank the reviewer for the questions. Blood glucagon is highest when the animal is fasting. Therefore, glucagon signaling is likely most active at this stage. At 7 dpf, the larva has exhausted the yolk and likely in a fasting state. Indeed, our previous study (Li Mingyu et al. Journal of Endocrinology, 2015, 227, 93-103), showed that 7 dpf appears to be an ideal time point for the study of α-cell hyperplasia phenotypes. That is why we use the 7 dpf larvae for the transcriptiome analysis.

Although there is no gross morphology phenotype, the liver shows already signs of steatosis and some of the changes in the transcriptomics profile they observe might be secondary effects and not the primary consequence of a GCGR knock out.

We totally agree that these changes in the gcgr mutant might be secondary effects. We add a sentence in the Discussion in the revised manuscript.

This should be further discussed and explicitely mentioned as a limitation of the current study (that would require different - earlier- timepoints of transcriptomic analyses to address)

We appreciate this criticism and added several sentences in the Discussion in the revised manuscript according to your suggestion.